# Effects of pesticide application on soil bacteria community structure in a cabbage-based agroecosystem in Ghana

Sefa Peprah[1]*, Patrick Addo-Fordjour[2], Bernard Fei-Baffoe[1], Kwadwo Boampong[2], Silas Wintuma Avicor[3], James Damsere-Derry[4]

**1** Department of Environmental Science, College of Science, Kwame Nkrumah University of Science and Technology, Kumasi, Ghana, **2** Department of Theoretical and Applied Biology, College of Science, Kwame Nkrumah University of Science and Technology, Kumasi, Ghana, **3** Entomology Division, Cocoa Research Institute of Ghana, New Tafo-Akim, **4** Council for Scientific and Industrial Research, Building Road and Research Institute, Ghana

* p.sefakus@gmail.com

## Abstract

Modern sustainable agriculture often relies on pesticide application, which may unintentionally affect non-target soil microorganisms. This study assessed the effects of commonly used pesticides in cabbage cultivation on bacteria diversity, composition, and abundance in soils from some farming communities in Bosome Freho District, Ghana. The pesticides included a neonicotinoid (acetamiprid), microbial agents (*Pieris rapae* granulosis virus+ *Bacillus thuringiensis*), avermectin (emamectin benzoate), and pyrrole (chlorfenapyr). Soil samples were collected from non-contaminated (NCS), abandoned pesticide-contaminated (AB-PCS) and actively pesticide-contaminated (AC-PCS) soils. Bacteria communities were analysed in the soil at phylum, class, order, family, genus, and species levels using 16S rRNA gene sequencing. The soils were also analyzed for physicochemical properties. Our results showed a decrease in bacteria diversity and abundance in pesticide-contaminated soils in the following order: NCS>AB-PCS>AC-PCS. Sorensen's coefficient of similarity indicated major shifts in bacteria taxa composition due to pesticide contamination. In NCS, *Pseudomonas veronii*, *Bacillus* sp., and *Prevotella albensis* were the most abundant species, while *Rhodoplanes elegans* and *Nostocoida limicola* dominated AB-PCS. In AC-PCS, *R. elegans*, *Gemmata obscuriglobus*, *Nitrospira calida*, and *N. limicola* were the most abundant species. The abundance of *Bacillus* sp., *P. veronii*, and *P. albensis* decreased in the contaminated soils, while the abundance of *N. calida*, *Cystobacter* sp., *Pedomicrobium australicum* and *Byssovorax cruenta* was higher in the contaminated soils. Key genera involved in nutrient cycling such as *Clostridium, Bacillus, Prevotella, Pseudomonas,* and *Arthrobacter*, declined in abundance in pesticide exposed soils. In contrast, an increase in abundance of various taxa such as *Pedomicrobium*, *Hyphomicrobiaceae*, *Pirellulaceae, Comamonadaceae,*

**Data availability statement:** All relevant data are within the manuscript and its supporting information file

**Funding:** The author(s) received no specific funding for this work.

**Competing interests:** The authors have declared that no competing interests exist.

*Nitrospirales*, *Nitrospira, Anaerolineae, Planctomycetes, Acidobacteriaí* and *Nitrospirae* was observed in the contaminated soils. These bacteria may possess bioremediation potential that could be exploited for environmental remediation. Soil physicochemical properties and nutrient levels varied across the three soil treatments, with potential implications for bacteria community structure.

## Introduction

Microorganisms play crucial roles in soil ecosystems and their activities are critical in nutrient composition, cycling and recycling [1–3]. Soils are perhaps the most complex and biodiverse ecosystems on earth containing nearly a quarter of the planet's diversity [4]. Soil biodiversity supports the provision of ecosystem services and these services are supported by soil properties such as organic carbon and organic matter content, nitrogen, and phosphorous among others [5–7]. Additionally, soil provides climate change mitigation services through carbon sequestration where soil organic matter acts as a carbon sink that mitigates the effects of greenhouse gases [8–10]. Soil bacteria are very important in regulating nutrient cycling and energy flow [11]. Some of these roles include decomposition of organic matter, and nitrogen fixation, which help maintain soil fertility [11,12]. However, unsustainable agricultural practices and the misuse of chemical pesticides are rapidly causing the depletion of soil fertility, threatening crop yield [13,14].

One of the major challenges in agriculture is frequent pest infestation, which tends to limit the growth and production of crops [14,15]. To overcome this challenge and meet the increased demand for food to feed the increasing population of the world, farmers resort to the use of pesticides to control pest populations during both pre- and post-harvest periods [14–16]. The use of pesticides has been widely adopted as an effective method for pest management and disease control [17]. Therefore, about one-third of agricultural produce are dependent on pesticide application [18]. Thus, as the entire world population increases rapidly, the demand for food would be expected to rise significantly, which may lead to a corresponding increase in the use of pesticides.

While the use of pesticides in agriculture is crucial to food security, extensive, indiscriminate, long-term and over-application of pesticides can adversely affect biodiversity and community structure of soil bacteria [13]. The applied pesticides get washed away from soils by storm water and thus find their way into water sources, leading to possible surface water contamination [19], which affects surface water quality and aquatic life [20]. Pesticides can also be emitted as volatile organic compounds through volatilization, which adversely affects air quality [21]. Another important impact of pesticide application lies in its influence on soil microbial community structure [22–25]. Pesticides can alter microbial diversity, abundance and composition, leading to shifts in ecosystem processes such as nutrient cycling [26]. Thus, despite their usefulness, residues of pesticides in the environment do not only cause loss of specific microbial population and functions, but are also detrimental to other organisms that inhale pesticide-contaminated air [14,27].

Cabbage (*Brassica oleracea*) is one of the most important vegetable crops grown almost throughout the year. In developing countries, pesticides are widely used in vegetable farming, especially in cabbage cultivation, in order to control pests and maximise yields. It has been reported that farmers apply copious amount of pesticides during cabbage cultivation, which can result in the contamination of the soil [28]. Nevertheless, there is limited research on how pesticide application affects soil bacterial communities in agroecosystems in developing countries such as Ghana. The potential impacts of pesticides on soil bacteria in agroecosystems are poorly understood. The lack of comprehensive studies on the long-term effects of pesticides on soil microbial diversity calls for the need for further research. Despite advancements in soil microbial research globally, microbiome assessments of agricultural soils in developing countries, especially Africa, using modern molecular techniques, such as high-throughput 16S rRNA gene amplicon sequencing and quantitative PCR, remain limited. The gene amplicon sequencing technique has several advantages over the traditional methods, including its superior sensitivity, resolution, and accuracy required for comprehensive bacterial community profiling [29]. In view of this, many previous studies which relied on conventional identification methods likely reported incomplete data on soil bacterial communities in relation to pesticide contamination.

The aim of this study was to investigate the effects of pesticides commonly used on cabbage fields on soil bacteria communities using gene amplicon sequencing. Given that pesticide application can affect soil bacteria, which in turn, can also influence soil properties, we also assessed key soil physicochemical properties across non-contaminated and pesticide-contaminated sites. Pesticides generally can cause negative effects on soil bacteria diversity and abundance due to the removal of sensitive species in the pesticide-treated soils [30]. Based on this, we hypothesized that pesticide application on cabbages will result in a reduction in bacteria diversity and abundance, and cause shifts in their composition. Furthermore, we anticipated that bacterial communities would recover in abandoned farms where pesticide application had ceased. Understanding the impact of pesticides on soil bacteria is important for developing sustainable agricultural practices. Thus, the findings of our study will provide important information on the broader implications of pesticide usage on soil bacteria communities. This information could guide the development of sustainable agricultural practices and inform policies aimed at minimising the negative impacts of pesticides on soil microorganisms.

## Materials and methods

### Study area

The study was conducted in Adeito and Adakabunso farming communities located in the northeastern part of the Bosome Freho District of the Ashanti Region in Ghana where cabbage is cultivated for commercial purposes. To gain access to the study area, the Head of Department of Environmental Science, KNUST obtained the site access permit by writing officially to the then Chief Executive Officer of the Bosome Freho District Assembly for approval.

The study community lies within the semi-deciduous forest zone in the north-eastern part of the district where large acreages of cocoa farms and other cash crops have been destroyed for cabbage cultivation. The topography of the area is characterized by an undulating landscape ranging between 240 and 300 meters above sea level drained by a few streams which take their sources from the highlands. However, the main water for irrigation in the study area is sourced from the Bankro stream which takes its source from the Adakabunso forest reserve and traverses through the Bosomtwe District, Juaben Municipal and Asante Akim South District of Ashanti Region [31]. The rainfall season of the district is bimodal, with the major season between March and July and the minor season between September and November. The mean annual rainfall is between 1,600 mm and 1,800 mm. The area experiences high and uniform temperatures ranging between 20°C and 30°C with moderate to high relative humidity between 70% and 80% during the rainy season [31]. The temperature regime and rainfall pattern enhance the cultivation of vegetables and cash crops throughout the year. The main soil type found in the study area is Ferric Acrisols [31] of which the texture is clay loamy. The soil is deep dark-brown clay, rich in nutrients and soil microorganisms, making the area a hub for the commercial cultivation of cabbage, cash crops and commercial trees such as cocoa, coffee and African mahogany [31].

The area is noted for commercial cabbage cultivation and as such commercial trees such as cocoa, orange planta-tions and part of the forest reserve have been destroyed for cabbage farming. According to farmers in the area, cabbage farming has been ongoing for over 30 years with continuous pesticides application. This has changed the vegetation of the crop land from semi-deciduous forest to high savanna with scattered tree stock. According to them, four different pesticides were used in the study belonging to the classes of neonicotinoids (acetamiprid), bacilli (*Bacillus thuringiensis*), avermectin (emamectin benzoate) and pyrrole (chlorfenapyr) with trade names as Golan 20 SL, Bypel 1, Attack 1.9 EC and Klopar respectively.

These pesticides have the optimal form of colloidal solutions which allow them to be applied by spraying with knapsack dispersion sprayer. The physicochemical properties of these pesticides show that they persist under simulated environ-mental conditions. For instance, acetamiprid has a half-life of approximately 31–450 days [32], Similarly, emamectin ben-zoate has aerobic soil metabolism half-life of 193 days [33], while chlorfenapyr has aerobic soil half-life of 1.4 years [34].

## Sampling procedure

Soil samples used for this study were collected from different locations in the Adeito-Adakabunso cabbage farming com-munity in January 2020. The farms were grouped according to previous history of pesticide application. This information was obtained from the farmers. The three main types of soils were non-contaminated, abandoned pesticide-contaminated and actively pesticide-contaminated. The actively pesticide-contaminated composite soil sample was taken from cabbage farms where pesticides use was still ongoing. The concentration of the applied pesticides was not considered in this study since an experimental farm was not set up in the study areas. However, the actual application rate used by the farmers indirectly provided information on the pattern of pesticide application rate in the area (Table 1). The abandoned pesticide-contaminated composite soil sample was also taken from fallow cabbage crop-fields where the same pesticides were used according to a survey (unpublished data) conducted in the study areas. We included the abandoned-contaminated soil sample in the study in order to evaluate the potential for bacterial community recovery over time. Finally, the non-contaminated soil sample was taken from an adjacent forest, where pesticides have not been applied.

Three sample sites were randomly selected in each sample location and quadrats of 5 m × 5 m were used to obtain five sub-samples from the corners and centers of the quadrats. Samples in each quadrat were mixed to form a com-posite sample from which sub-samples were taken. Samples were taken using trowels from 20 cm square at a depth of 20 cm where biomass is typically abundant. Triplicate samples were collected within each quadrat to a minimum volume of 0.5 kg. The trowels used were disinfected with 70% ethanol and rinsed with sterile water before the next soil sample collection. Samples were collected in tightly sealed plastic bags and kept at 4°C to keep the field moist and preserve biological properties. Samples were transported in ice packs to the Department of Theoretical and Applied Biology Labora-tory, at the Kwame Nkrumah University of Science and Technology (KNUST) for analysis. In the laboratory, samples were analyzed less than 24 hours after sample collection to avoid DNA degradation.

**Table 1. List of pesticides used on cabbage fields in the study area. WHO hazard class II = moderately hazardous, *class* III = slightly hazardous.**

| Trade Name/ Product | Active Ingredient/ Concentration | Chemical Group | WHO Haz-ard Class | Target Pest | Applica-tion Rate |
|---|---|---|---|---|---|
| Attack 1.9 EC | Emamectin benzoate 1.9% w/w EC | Avermectin | II | Control of insect pests in vegetables | 10 mL/15 L |
| Bypel 1 | Pierisrapae Granulosis Virus 10,000PIB/mg + *Bacillus thuringiensis* 16,000IU/mg | Bacilli (Bio-insecticide/ Microbial insecticide) | II | Control of white flies and worms in vegetables and fruits | 10 g/15 L |
| Golan 20 SL | Acetamiprid 200g/L | Neonicotinoid | III | Control of insect pests in vegetables, citrus, cotton, coffee and maize | 10 mL/ 15L |
| Klopar 24 SC | Chlorfenapyr 1.5% | Pyrrole | II | Control of insect pests in vegetables | 10 mL/15 L |

## DNA extraction and 16S rRNA gene sequencing

To identify the taxonomic diversity and composition of bacteria in the soil, genomic DNA was extracted from 10 g of each of the soil replicates using the Zymo Quick-DNA Faecal/Soil Microbe Microprep kit (Inqaba Biotec, South Africa) according to the manufacturer's protocol [35]. The concentration and purity of the extracted DNA was determined using a NanoDrop spectrophotometer at 260 nm and 280 nm. The DNA quality was assessed by agarose gel electrophoresis. The extracted DNA samples were normalized, and equal amounts were analysed by 16S ribosomal RNA (rRNA) gene sequencing. Briefly, the genomic DNA samples were PCR amplified using a universal primer pair 27F and 1492R – targeting the V1 – V9 region of the bacterial 16S rRNA gene. The resulting amplicons were barcoded with PacBio M13 barcodes for multiplexing through limited cycle PCR. The resulting barcoded amplicons were quantified and pooled equimolar and AMPure PB bead-based purification step was performed. The PacBio SMARTbell library was prepared from the pooled amplicons following manufacture protocol. Raw subreads were processed through SMRTlink (v7.0.1). These highly accurate reads were then processed through vsearch tool (30). Taxonomic information was determined based on QIMME2.

## Determination of soil chemical characteristics

Soil samples were analysed to assess the levels of pH, organic matter, total nitrogen (N), available phosphorus (P), total potassium (K), total calcium (Ca), total magnesium (Mg), total sodium (Na), and cation exchange capacity (CEC). Soil pH was determined using a pH meter (Jenway 3305) in 1:1 soil:water ratio. Organic matter contents were determined using the Walkley-Black method [36]. Total N was also measured using the Kjeldahl method [37]. We extracted available P using the Bray I and Bray II method, while its concentration was quantified by spectrophotometry [38]. We determined the content of exchangeable K, Ca, Mg, and Na by acid-digesting the soil samples. We then subjected the resulting solution to flame atomic absorption spectrophotometry (Perkin Elmer Analyst 100) to determine the concentrations of the cations [39]. The cation exchange capacity (CEC) was determined using the ammonium acetate exchange method described by [40]. The proportions of sand, silt, and clay in the soils were determined using the hydrometer method [41], and the values obtained were used to classify the soils.

## Data analysis

Metagenomics sequence data obtained from Inqaba Biotech™ (Pretoria, South Africa) were processed and edited using the basic local alignment search tool (ElasticBLAST version 1.0.0). The data were normalized using the Relative Log Expression to facilitate comparisons as the number of reads was comparable among various soil treatments. Subsequently, the normalized data was analyzed using Microsoft Excel to get the read count.

We used the PAST statistical package to quantify Fisher's alpha diversity and compare it among the three soil treatments for each of the various bacteria taxa using permutation tests. The richness of the different bacteria taxa was also compared among the different soil treatments using permutation tests of PAST. To determine the patterns of bacteria taxa composition among all the pairs of sites denoting the three soil treatments, we calculated Sorenson's coefficient using the following equation:

$$\text{Sorenson's Coefficient} = \frac{2C}{S1 + S2}$$

Where, C is the number of species the two sites have in common, S1 is the total number of species in site 1, and S2 is the total number of species in site 2

The bacteria load in the various taxa were compared among the three soil treatments by running one-way ANOVA in STATA version 13 (Collage Station, Texas 77845 USA).

## Results

### Taxonomic diversity, composition, and abundance of soil bacteria in various soils

**Bacteria species in soils.** The non-contaminated soil supported a significantly higher bacteria species richness than both pesticide-contaminated soils. However, the abandoned and actively pesticide-contaminated soils harboured comparable species richness of bacteria (P = 0.632) (Table 2; S1 Table). Fisher's alpha diversity of bacteria species was similar between the non-contaminated and actively pesticide-contaminated soils, but each of them was significantly lower than that in the abandoned pesticide-contaminated soil (Table 2). Moreover, Sorensen's coefficient of similarity showed that there was a very low similarity in the bacterial species composition between the non-contaminated soils and the abandoned pesticide-contaminated soils, as well as between the abandoned and actively pesticide-contaminated soils (Table 3). However, there was moderate similarity in bacterial genus and species composition between the non-contaminated and actively pesticide-contaminated soils. The non-contaminated soils had low species similarity with the abandoned pesticide-contaminated soils. Likewise, there was moderate similarity in the composition of bacterial family and species between the non-contaminated and abandoned pesticide-contaminated soils.

The total bacteria species load detected in the non-contaminated soils was 10,357 sequences, with 71.03% belonging to unknown species groups (S1 Table). The most abundant species in the non-contaminated soils were *Pseudomonas venorri*, *Bacillus* sp., *Prevotella albensis* and *Bacillus flexus* (Table 4). Similarly, species detected in the abandoned pesticide-contaminated soils generated 4,223 sequences, where 85.58% belonged to unknown species. *Rhodoplanes elegans*, *Nostocoida limicola* and *Gemmata obscuriglobus* were the most abundant known species in the abandoned pesticide-contaminated soils. In the actively pesticide-contaminated soil samples, 1,644 sequences were detected, with *Rhodoplanes elegans*, *Gemmata obscuriglobus*, *Nitrospira calida* and *Nostocoida limicola* being the most abundant species. The unknown species detected accounted for 74.39% of the species abundance. The chi-square results revealed significant differences in the abundance of bacteria species between the three soil treatments (P = 0.001) (Table 5). Generally, most species recorded the greatest abundance in the non-contaminated soil.

**Bacteria genus in soils.** We observed a significantly higher number of bacteria genera in the non-contaminated soil than in the abandoned pesticide-contaminated soils at the genus level (Table 2; S2 Table). Similarly, the non-contaminated soils supported a significantly higher number of bacteria genera than the actively pesticide-contaminated soil. Furthermore, the number of bacteria genera in the actively pesticide-contaminated soil was significantly lower than the abandoned pesticide-contaminated soil. There were no significant differences in Fisher's alpha diversity of bacteria genera among the different soil treatments (Table 2). The genus composition similarity was low among all the sites, except for a moderate similarity between the non-contaminated and actively pesticide-contaminated soils (Table 3).

**Table 2. A comparison of species diversity of various bacteria taxa in non-contaminated soils (NCS), abandoned pesticide-contaminated soil (AB-PCS) and actively pesticide-contaminated soil (AC-PCS). For each index, values in the same row having different superscript letters are significantly different, as determined by permutation tests.**

| | Species richness | | | | Fisher alpha diversity | | |
|---|---|---|---|---|---|---|---|
| Taxa | NCS | | AB-PCS | AC-PCS | NCS | AB-PCS | AC-PCS |
| Species | 110[a] | | 99[b] | 64[b] | 22.43[a] | 33.52[b] | 22.44[b] |
| Genus | 75[a] | | 61[b] | 52[c] | 13.92[a] | 15.62[a] | 13.16[a] |
| Family | 129[a] | | 121[a] | 103[b] | 20.26[a] | 26.00[b] | 22.86[a] |
| Order | 56[a] | | 48[b] | 37[c] | 7.44[a] | 9.26[b] | 6.71[a] |
| Class | 33[a] | | 31[a] | 23[b] | 4.13[a] | 4.95[a] | 3.47[b] |
| Phylum | 20[a] | | 15[a] | 14[a] | 2.28[a] | 1.90[a] | 1.83[a] |

**Table 3. Soil bacteria composition similarity in the non-contaminated (NCS), abandoned pesticide-contaminated (AB-PCS) and actively pesticide-contaminated (AC-PCS) soils.**

| Taxa | Sorensen's coefficient | |
|---|---|---|
| | NCS | AB-PCS |
| | **Phylum** | |
| NCS | – | – |
| AB-PCS | 0.04 | |
| AC-PCS | 0.30 | 0.34 |
| | **Class** | |
| NCS | – | – |
| AB-PCS | 0.11 | |
| AC-PCS | 0.21 | 0.35 |
| | **Order** | |
| NCS | – | – |
| AB-PCS | 0.17 | |
| AC-PCS | 0.10 | 0.26 |
| | **Family** | |
| NCS | – | – |
| AB-PCS | 0.11 | |
| AC-PCS | 0.33 | 0.20 |
| | **Genus** | |
| NCS | – | – |
| AB-PCS | 0.36 | |
| AC-PCS | 0.58 | 0.24 |
| | **Species** | |
| NCS | – | – |
| AB-PCS | 0.20 | |
| AC-PCS | 0.70 | 0.34 |

The total bacteria load in the non-contaminated samples at the genus level was 9,100 sequences with 64.22% belonging to unknown genera (S2 Table). *Bacillus, Pseudomonas* and *Prevotella* were the most abundant genera (Table 4). Similarly, the abandoned pesticide-contaminated soil samples at the genus level produced 4,410 sequences, with 81.61% of the bacteria sequences belonging to unknown genera (S2 Table). In the actively pesticide-contaminated soils, 3,927 sequences were detected, out of which 81.54% belonged to unknown genera. Among the known bacteria genera in both pesticide-contaminated soils, *Rhodoplanes, Gemmata* and *Nostocoida* were the most abundant genera in the abandoned pesticide-contaminated soil.

**Bacteria families in soils.** The number of bacteria families present in the non-contaminated and abandoned pesticide-contaminated soils were similar. Nevertheless, the actively pesticide-contaminated soil supported a significantly lower family richness than the non-contaminated and abandoned pesticide-contaminated soils (P = 0.001) (Table 2; S3 Table). There were no significant differences in Fisher's alpha diversity between the non-contaminated and actively pesticide-contaminated soils, although each of them contained a significantly lower Fisher's alpha diversity of bacteria family than the abandoned pesticide-contaminated soil. We recorded a low family composition similarity among all the sites, with a slightly higher similarity between the non-contaminated and actively pesticide-contaminated soils compared to the abandoned and actively pesticide-contaminated soils (Table 3).

The total bacteria family load detected in the non-contaminated soils was 16,074 sequences, with the most abundant families being *Veillonellaceae, Micrococcaceae* and *Bacillaceae* (Table 4; S3 Table). Bacteria belonging to unknown

**Table 4. Relative abundance (%) of the topmost bacteria taxa in the taxonomic hierarchy within the non-contaminated (NCS), abandoned pesticide-contaminated (AB-PCS) and actively pesticide-contaminated (AC-PCS) soils.**

| Taxa | Treatment | | |
|---|---|---|---|
| | NCS | AB-PCS | AC-PCS |
| **Species Level** | | | |
| *Bacillus* sp. | 5.21 | – | – |
| *Nitrospira calida* | – | – | 2.13 |
| *Prevotella albensis* | 3.58 | – | – |
| *Pseudomonas veroni* | 6.23 | – | – |
| *Rhodoplanes elegans* | – | 1.52 | 5.05 |
| *Nostocoida limicola* | – | 1.14 | – |
| *Gemmata obscuriglobus* | – | 0.90 | 2.13 |
| *Bacillus flexus* | 1.98 | – | – |
| **Genus Level** | | | |
| *Bacillus* | 9.88 | – | – |
| *Gemmata* | – | 2.34 | 2.55 |
| *Prevotella* | 5.33 | – | – |
| *Nostocoida* | – | 2.11 | 1.76 |
| *Pseudomonas* | 6.25 | – | – |
| *Rhodoplanes* | – | 2.45 | 4.53 |
| **Family Level** | | | |
| *Bacillaceae* | 9.67 | – | – |
| *Gemmataceae* | 8.55 | 14.79 | 15.09 |
| *Hyphomicrobiaceae* | – | 3.52 | 5.27 |
| *Lachnospiraceae* | 3.17 | – | – |
| *Micrococcaceae* | 10.87 | – | – |
| *Pirellulaceae* | – | 2.06 | 1.94 |
| *Pseudomonadaceae* | 5.75 | – | – |
| *Prevotellaceae* | 5.42 | – | – |
| *Veillonellaceae* | 11.04 | – | – |
| **Order Level** | | | |
| *Actinomycetales* | 15.03 | 3.87 | 5.21 |
| *Bacillales* | 13.19 | 2.67 | – |
| *Bacteroidales* | 15.93 | – | – |
| *Burkholderiales* | 3.70 | 3.52 | 2.00 |
| *Clostridiales* | 22.53 | – | – |
| *Gemmatales* | 3.97 | 18.32 | 19.35 |
| *Myxococales* | – | 3.78 | 3.47 |
| *Nitrospirales* | – | 3.70 | 8.57 |
| *Rhizobiales* | 3.11 | 14.24 | 30.13 |
| *Rhodospirillales* | – | – | 2.10 |
| **Class level** | | | |
| *Clostridia* | 22.73 | – | – |
| *Actinobacteria* | 14.98 | 2.94 | 4.00 |
| *Alphaproteobacteria* | 4.62 | 17.50 | 29.45 |
| *Bacilli* | 13.25 | 8.32 | 2.59 |
| *Gamaproteobacteria* | 6.80 | – | 10.61 |
| *Acidobacteriia* | – | 6.49 | 6.25 |

*(Continued)*

**Table 4.** (Continued)

| Taxa | Treatment | | |
|---|---|---|---|
| | NCS | AB-PCS | AC-PCS |
| *Planctomycetia* | 4.44 | 15.93 | 10.78 |
| *Betaproteobacteria* | – | 11.74 | – |
| *Nitrospira* | – | 3.52 | 9.74 |
| **Phylum level** | | | |
| *Acidobacteria* | 2.44 | 11.63 | 11.62 |
| *Actinobacteria* | 17.71 | 4.87 | 8.89 |
| *Bacteroidetes* | 16.57 | – | – |
| *Firmicutes* | 35.50 | 3.86 | 3.17 |
| *Planctomycetes* | 4.78 | 21.44 | 16.62 |
| *Proteobacteria* | 17.73 | 40.34 | 37.13 |
| *Nitrospirae* | – | 3.05 | 5.00 |

families accounted for 26.78% of the total sequences. In the abandoned pesticide-contaminated soils, a total of 6,614 sequences were generated, with 59.12% belonging to unknown families. The most abundant families were *Gemmataceae, Hyphomicrobiaceae* and *Pirellulaceae.* The actively pesticide-contaminated soils also generated 4,645 sequences, out of which 55.93% were for unknown families. The most abundant bacteria families identified in the pesticide-contaminated soils were *Gemmataceae* and *Hyphomicrobiaceae* and *Pirellulaceae* (S3 Table).

**Bacteria orders in soils.** Generally, the number of bacteria orders decreased with pesticide application according to the following order: non-contaminated > abandoned pesticide-contaminated > actively pesticide contaminated. There were significant differences in the bacteria order richness among the three types of soil treatments (P = 0.001) (Table 2; S4 Table). Furthermore, the abandoned pesticide-contaminated soil harboured significantly higher Fisher's alpha diversity than the non-contaminated and actively pesticide-contaminated soils. However, the Fisher's alpha diversity in the non-contaminated soil was similar to that in the actively pesticide-contaminated soil. The similarity in bacteria order composition among all the three sites was low (Table 3).

The order level generated 15,039 sequences in the non-contaminated soil, with the most abundant bacteria orders being *Clostridiales, Bacteroidales, Actinomycetales* and *Bacillales* (Table 4; S4 Table). Within the abandoned pesticide-contaminated soils, 2,325 sequences were identified, and two orders namely, *Gemmatales* and *Rhizobiales* were the most abundant. Similarly, there were 1,902 sequences of bacteria order in the actively pesticide-contaminated samples, in which *Rhizobiales, Gemmatales* and *Nitrospirales* constituted the most abundant bacteria orders in the soil.

**Bacteria classes in soils.** Both the non-contaminated and abandoned pesticide-contaminated soils supported comparable numbers of bacteria class and Fisher's alpha diversity at the class level (P = 0.964) (Table 2; S5 Table). Nonetheless, there were significantly higher richness and Fisher's alpha diversity of bacteria class between the abandoned and actively pesticide-contaminated soils (P = 0.037). Similarly, the non-contaminated soil harboured significantly higher number of bacteria classes and class Fisher's alpha diversity than the actively pesticide-contaminated soils (P = 0.001). There was a low similarity in bacteria composition between the non-contaminated and abandoned pesticide-contaminated soils as well as between the abandoned and actively pesticide-contaminated soils (Table 3). However, a higher similarity in bacteria class composition was observed between the non-contaminated and actively pesticide-contaminated soils.

A total of 12,942 counts of bacteria class were detected in the non-contaminated soils, out of which 5.02% were unknown (S5 Table). The most abundant bacteria classes in the non-contaminated soils were *Clostridia, Actinobacteria*, and *Bacilli* (Table 4). We counted a total number of 3,126 individuals of bacteria classes in the abandoned

pesticide-contaminated soil (S5 Table). About 16.63% of the class abundance in the abandoned pesticide-contaminated soil belonged to unknown classes. The most abundant classes in the abandoned pesticide-contaminated soil were *Alphaproteobacteria, Planctomycetia* and *Betaproteobacteria*. In the case of the actively pesticide-contaminated soil, we counted a total of 2,978 sequences, out of which 10.98% belonged to unknown classes (S5 Table). The bacteria classes with the highest abundance in the actively pesticide-contaminated soil were *Alphaproteobacteria, Planctomycetia* and *Gammaproteobacteria.*

**Bacteria phyla in soils.** We did not observe significant differences in the richness and Fisher's alpha diversity of bacteria phyla among the three soil treatments (Table 2; S6 Table). The composition of the bacteria phyla was found to be very low between the non-contaminated and abandoned pesticide-contaminated soils, and between abandoned and actively pesticide-contaminated soils (Table 3). In contrast, a moderate similarity was observed in the composition of bacteria phyla between the non-contaminated and actively pesticide-contaminated soils.

The highest number of bacteria phyla load was counted for the non-contaminated soil (9,860 sequences), followed by the abandoned pesticide-contaminated soil (3,344 sequences), and the active pesticide-contaminated soil (2,521 sequences). Out of these numbers, only a few of the sequences belonged to unidentified phyla (non-contaminated soil: 2.24%; abandoned pesticide-contaminated soil: 9.15%; actively pesticide-contaminated soil: 10.67% (S6 Table). The most abundant phyla in the non-contaminated soils were *Firmicutes, Proteobacteria, Actinobacteria* and *Bacteroidetes,* while *Proteobacteria, Planctomycetes* and *Acidobacteria* were the most abundant phyla in the abandoned pesticide-contaminated soils (Table 4). In the actively pesticide-contaminated soils, *Proteobacteria, Planctomycetes* and *Acidobacteria* formed the most abundant phyla. There were significant differences in bacteria phyla abundance among the three soil treatment types, with most of the phyla having the greatest abundance in the non-contaminated soils (P = 0.001)

**Table 5. Comparison of the abundance of the topmost twenty bacteria species in the non-contaminated soil (NCS), abandoned contaminated soil (AB-PCS) and actively contaminated soil (AC-PCS) in the study area.**

| Species | NCS | AB-PCS | AC-PCS | P-value |
|---|---|---|---|---|
| *Bacillus* sp. | 540 | 18 | 12 | 0.001 |
| *Pseudomonas veronii* | 645 | 2 | 0 | 0.001 |
| *Prevotella albensis* | 371 | 2 | 0 | 0.001 |
| *Prevotella ruminicola* | 184 | 16 | 8 | 0.001 |
| *Selenomonas lacticifex* | 68 | 2 | 6 | 0.001 |
| *Rhodoplanes elegans* | 88 | 64 | 83 | 0.001 |
| *Gemmata obscuriglobus* | 33 | 38 | 35 | 0.001 |
| *Nostocoida limicola* | 55 | 48 | 25 | 0.001 |
| *Bacillus koreensis* | 29 | 2 | 1 | 0.001 |
| *Arthrobacter woluwensis* | 28 | 2 | 6 | 0.001 |
| *Alicyclobacillus acidiphilus* | 26 | 2 | 1 | 0.001 |
| *Paenibacillus curdianolyticus* | 26 | 6 | 0 | 0.001 |
| *Clostridium butyricum* | 22 | 2 | 0 | 0.001 |
| *Gemmata eligans* | 32 | 16 | 12 | 0.001 |
| *Singulisphaera rosea* | 20 | 10 | 10 | 0.001 |
| *Eggerthella sinensis_* | 21 | 1 | 0 | 0.001 |
| *Pseudomonas carboxydohydrogena* | 16 | 20 | 14 | 0.001 |
| *Nitrospira calida* | 9 | 28 | 35 | 0.001 |
| *Planctomyces* | 25 | 28 | 8 | 0.001 |
| *Pedomicrobium australicum* | 12 | 26 | 9 | 0.001 |

(Table 6). The abundance of most of the bacteria phyla decreased in the pesticide-treated soils, except for some phyla such as *Nitrospirae*, *Armatimonadetes*, *Planctomycetes* and *Acidobacteria*.

## Soil nutrients and chemical properties

Soil pH in the non-contaminated and actively pesticide-contaminated soils were strongly acidic, while that of abandoned pesticide-contaminated soil was slightly acidic (Table 7). The organic matter content was highest in the actively pesticide-contaminated soil, followed by the abandoned pesticide-contaminated soil, and then the non-contaminated soil. Meanwhile, total N contents were similar in the non-contaminated soil, and the abandoned and actively pesticide-contaminated soils. Available P concentration in the abandoned pesticide-contaminated soil was approximately seven times higher than in the non-contaminated soil, while the available P level in the actively pesticide-contaminated soil was about eight times higher than in non-contaminated soils. Similarly, the non-contaminated soil contained lower levels of K, Ca, Mg, and Na than the abandoned and actively pesticide-contaminated soils. Furthermore, cation exchange capacity (CEC) was lower in the non-contaminated soil compared to the two types of pesticide-contaminated soils. The three soils were generally clayey with the non-contaminated soil and the abandoned pesticide-contaminated soil having clay loam texture, while the actively contaminated soil had a clay texture.

## Discussion

### Soil bacteria diversity, composition and abundance

In this study, we investigated the effects of pesticides commonly used in cabbage cultivation on bacteria community structure measured by species diversity, composition, and abundance. Soil bacteria are crucial in maintaining soil health and fertility through nutrient cycling and their antagonistic effects against soil pathogens [1–3]. Previous studies showed that pesticide application decreased diversity and abundance of soil bacteria by inducing stress conditions on them [42]. Thus, following a decline in bacteria diversity, there could be disruptions in ecosystem services, with ramifications for plant growth and soil nutrient availability. At all taxonomic levels, we observed shifts in bacteria composition in the abandoned pesticide-contaminated and actively pesticide-contaminated soils in relation to the non-contaminated soil. This phenomenon may be related to pesticide application, which tends to modify bacteria community structure as reported in previous studies [30,43]. While some bacterial taxa are sensitive to pesticides and decline in abundance [22,23], others thrive by utilising pesticides as a carbon source [24,25], resulting in shifts in microbial composition in pesticide-contaminated soils. These dynamics may lead to reduced ecosystem resilience, as the loss of pesticide-sensitive taxa may negatively affect essential soil processes and interactions. We also noticed that bacteria abundance decreased from the non-contaminated soil (10,357 sequences) through the abandoned pesticide-contaminated soil (4,223 sequences) to the actively pesticide-contaminated soil (1,644 sequences). This trend indicates a significant decline in bacteria abundance with increasing pesticide contamination. The sharp decline in the bacterial numbers in the actively pesticide-contaminated soils suggests that ongoing pesticide application exerts a strong selective pressure on the microbial communities [26]. The lower bacteria population size in the contaminated soils may be attributed to toxic effects of pesticides, which could selectively inhibit or eliminate sensitive bacteria taxa, but favour resistant ones [24,25,44,45].

### Effects of pesticide application on bacteria species

At the species level, the non-contaminated soil was dominated by *Pseudomonas veronii, Bacillus* sp*., Prevotella albensis* and *Bacillus flexus. Among these, *Pseudomonas* and *Bacilllus* spp. are widely abundant in soils [46] and play important roles in bioremediation [47]. These bacteria species exhibit remarkable metabolic adaptability, enabling them to degrade a wide range of pesticides and aromatic compounds [48,49]. This phenomenon enables them to dominate in pesticide-contaminated soils, where they can contribute considerably to pollutant degradation and soil restoration. However, our

**Table 6. Comparison of the abundance of the topmost 16 bacteria phyla in the non-contaminated soil (NCS), abandoned contaminated soil (AB-PCS) and actively contaminated soil (AC-PCS) in the study area.**

| Phylum | NCS | AB-PCS | AC-PCS | P-value |
|---|---|---|---|---|
| *Firmicutes* | 3500 | 129 | 80 | 0.001 |
| *Actinobacteria* | 1746 | 163 | 224 | 0.001 |
| *Proteobacteria* | 1748 | 1349 | 936 | 0.001 |
| *Bacteroidetes* | 1634 | 43 | 31 | 0.001 |
| *Planctomycetes* | 471 | 717 | 419 | 0.001 |
| *Acidobacteria* | 241 | 389 | 293 | 0.001 |
| *Chloroflexi* | 48 | 68 | 45 | 0.001 |
| *Nitrospirae* | 49 | 102 | 126 | 0.001 |
| *Gemmatimonadetes* | 36 | 34 | 34 | 0.001 |
| *Lentisphaerae* | 38 | 2 | 0 | 0.001 |
| *Cyanobacteria* | 25 | 11 | 6 | 0.001 |
| *Spirochaetes* | 24 | 4 | 0 | 0.001 |
| *Verrucomicrobia* | 21 | 15 | 12 | 0.001 |
| *Armatimonadetes* | 6 | 0 | 23 | 0.001 |
| *Synergistetes* | 29 | 0 | 6 | 0.001 |
| *Tenericutes* | 5 | 0 | 0 | 0.001 |

**Table 7. Chemical properties of non-contaminated (NCS), abandoned pesticide-contaminated (AB-PCS) and actively pesticide-contaminated (AC-PCS) soils.**

| Soil property | NCS | AB-PCS | AC-PCS |
|---|---|---|---|
| pH | 5.14±0.16 | 6.69±0.09 | 4.93±0.17 |
| Organic matter (%) | 3.63±0.33 | 4.00±0.70 | 4.7±0.15 |
| N (%) | 0.18±0.02 | 0.20±0.01 | 0.2±0.02 |
| P (mg/kg) | 3.56±0.30 | 25.62±1.02 | 30.26±1.26 |
| K (mg/kg) | 0.15±0.01 | 0.49±0.07 | 0.30±0.03 |
| Ca (mg/kg) | 3.62±0.12 | 9.8±0.10 | 8.31±0.21 |
| Mg (mg/kg) | 2.77±0.17 | 4.47±0.27 | 5.33±0.13 |
| Na (mg/kg) | 0.07±0.01 | 0.13±0.01 | 0.14±0.02 |
| CEC (meq/100g) | 6.61±0.21 | 14.89±0.11 | 14.37±0.37 |

study revealed that the abundance of *P. veronii* and *Bacillus* sp. was significantly lower in the pesticide-contaminated soils compared to the non-contaminated soils. The extremely low abundance of these species in the pesticide-contaminated soils suggests that pesticide application might have had adverse effects on the populations of the species. In this regard, our finding is at variance with that reported in a previous study in which higher abundance of *Pseudomonas* and *Bacillus* spp. were recorded in pesticide-contaminated soils [42]. These results suggest that the specific pesticides present in our pesticide-contaminated soils may not belong to the form that can be metabolised by these bacteria. Pesticide toxicity could be responsible for possible reduction in the abundance of these species in the pesticide-contaminated soil [45]. If this is the case, then the pesticide-degradation property reported for these bacteria is not universal but limited to specific pesticides.

The low abundance of *P. veronii* and *Bacillus* sp. in the pesticide-contaminated soils may negatively affect soil fertility and the overall productivity of the cabbage field soils. These species possess catabolic pathways and enzymes that enable them to degrade pesticides [42,50]. *Pseudomonas* spp., as facultative aerobes, play a crucial role in nitrogen

cycling, degradation of aromatic compounds, and plant growth promotion [51,52]. In the same vein, *Bacillus* spp. contribute to soil fertility by converting complex nutrients like phosphate and nitrate into simpler forms for plant uptake [53]. Additionally, both bacteria species help protect plants from diseases by forming biofilms and secreting toxins that suppress pathogenic bacteria [54]. Given their ecological importance, their low numbers in the pesticide-contaminated soils could have adverse consequences for soil fertility.

## Effects of pesticide application on bacteria genera

At the genus level, *Bacillus*, *Gemmata*, *Prevotella, Nostocoida, Pseudomonas*, and *Rhodoplanes* were the most abundant in the soil. Within the actively pesticide-contaminated soil, *Rhodoplanes* occurred as one of the most abundant genera, suggesting that bacteria in this genus may play a role in pesticide degradation in soils. This group of bacteria may utilise pesticides in the soil as a carbon source, enabling them to thrive in the pesticide-contaminated soil as reported for other bacteria taxa [24,25]. This phototrophic genus includes strains that possess nitrogen-fixing capabilities, which allow it to convert atmospheric nitrogen into a usable form [55]. Therefore, the *Rhodoplanes* in our soils may contribute to soil nitrogen enrichment. Our findings demonstrated a decrease in the diversity and composition of some bacteria genera in areas exposed to pesticides. However, the persistence of certain bacterial genera despite pesticide exposure suggests that these bacteria are promising candidates for bioremediation. *Pseudomonas* is reported as a diverse genus with catabolic pathways and enzymes necessary for pesticide degradation [42]. Pesticide-degradating bacteria such as *Pesudomonas* often exhibit higher population size in contaminated soils, as they can utilise pesticides as carbon or energy source [56]. Contrary to this expectation, we observed a higher abundance of *Pseudomonas* in the non-continted soil. a higher abundance of this genus in the pesticide-contaminated soils, showing that abundance alone may not be a reliable indicator of its role in pesticide degration. Further studies, such as functional assays, are needed to confirm its potential role in pesticide degradation.

## Effects of pesticide application on bacteria families, orders and classes

Our findings revealed that *Gemmataceae* dominated in both the non-contaminated and contaminated soils. The presence of *Gemmataceae* in both environments suggests that this bacterial group is ubiquitous and robust. Previous studies have identified *Gemmataceae* in diverse environments such as swamps, wastewater treatment plants, peatlands, and acid bogs [57]. These findings, combined with our results, indicate that *Gemmataceae* may possess resistance mechanisms against environmental stressors that enable them to thrive under different environmental stressors. Interestingly, *Gemmataceae* also thrives in nutrient-rich soils, which may explain its higher abundance in the non-contaminated soil.

Previous studies demonstrated that bacterial orders such as *Xanthomonadales, Sphingomonadales* and *Pseudomonadales* degrade chlorinated pesticides, while others such as *Burkholderiales, Rhizobiales* and *Acidobacteriales* are considered potential pesticide degraders due to their high abundance in contaminated soils [58–60]. However, in our study, most of these bacterial orders appear to have exhibited a decline in abundance in pesticide-contaminated soils, thus contrasting with previous findings. The conflicting findings between our study and previous ones suggest that the degradation ability and abundance of these orders may be influenced by local environmental factors such as soil composition, and pesticide type and concentration. Thus, their pesticide-degradation activities may not follow universal patterns. However, the abundance of *Nitrospirales* and *Rhizobiales* was higher in at least one of the pesticide-contaminated soils. Bacteria species in these orders may be active metabolisers of pesticides, or they may possess mechanisms that allow them to tolerate pesticides. The high abundance of these bacterial orders in the pesticide-contaminated soils may be beneficial for agriculture, as they are involved in nutrient cycling. For example, *Nitrospirales* oxidizes nitrite and fix carbon dioxide [61], while *Rhizobiales* fix nitrogen in the soil [62].

In both pesticide-contaminated soils, we identified *Alphaproteobacteria* and *Planctomycetia* as the most abundant classes. This finding is consistent with previous studies which observed that *Alphaproteobacteria* was among the most

abundant bacteria classes in their samples [63,64]. Moreover, our study also identified *Gammaproteobacteria* and *Betaproteobacteria* among the most abundant classes in the pesticide-contaminated soils, a trend which is consistent with Malla et al. (2022). In contrast, the most abundant bacteria classes identified in the non-contaminated soil differed from those found in the contaminated soils. This pattern may indicate a shift in bacteria class composition due to pesticide exposure.

## Effects of pesticide application on bacteria phyla

The taxonomic structure of microbial community at the phylum level was in general quite typical of microbial communities in the rhizosphere soils. We observed the predominance of *Firmicutes, Actinobacteria, Proteobacteria, Bacteroidetes, Planctomycetes* and *Acidobacteria* in our study area. The predominance of these bacteria and others observed in our study area are consistent with previous studies which identified these groups as common inhabitants in rhizosphere soils [65,66]. The abundance of *Nitrospirae and Planctomycetes* increased in the pesticide-contaminated soils. These groups are among the important bacteria groups that degrade diverse groups of pesticides and inorganic compounds [67]. This finding is similar to previous studies that also found these groups of bacteria in high abundance in pesticide-contaminated soils [68]. The high abundance of *Planctomycetes* and *Nitrospirae* in the pesticide-contaminated soils could be associated with their ability to use pesticides as food and energy sources [56]. A number of related studies demonstrated that the abundance of *Actinobacteria* decreased in pesticide-contaminated soils [22,23]. The differences in the responses of these phyla to pesticides in the soils suggest that soil different bacteria taxa may exhibit varied responses to specific pesticides. This brings to the fore the importance of conducting site-specific studies of pesticide effects on soil bacterial communities for a better understanding of their ecological impact. By being subjected to farming for several years, the pesticide-treated soils were expected to have lower soil N due to potential depletion by crops. However, N concentration in the two pesticide-treated soil treatments was comparable to that in the non-treated soil. This phenomenon suggests that other mechanisms may be maintaining soil N in the pesticide-treated soils. Generally, members of *Actinobacteria* exhibit several economic and agricultural relevance [69]. Our results showed that *Actinobacteria* was less abundant in the pesticide-treated soils than the non-treated soils. Given that this phylum plays a very important role in soil breakdown, humus construction, and nitrogen fixation in the soil ecosystem [70], their lower population size may affect nitrogen build up in the pesticide-contaminated soils.

## Soil properties and bacteria community

Both macro- and micro-nutrients as well as ECEC exhibited higher levels in the pesticide-contaminated soils compared to the non-contaminated soils. While these variations in soil properties likely influenced microbial community structure, it is also possible that bacterial communities played a key role in determining the soil properties. Some bacteria taxa contribute to key soil process such as nitrogen cycling, phosphorus solubilisation, and organic matter decomposition. For instance, species from bacteria orders, *Rhizobiales* and *Nitrospirales* are important in nitrogen and carbon cycling, and can therefore influence soil nitrogen and organic carbon contents [61,62]. Moreover, specific nutrient-solubilising bacteria in the pesticide-contaminated soils may have facilitated the release of bioavailable nutrients, leading to the observed higher levels of soil P, K, Mg, Ca, and Na.

## Limitations of the study

Although our study demonstrated the effects of pesticide application on soil bacteria community structure, the reliance on farmers' self-reported pesticide usage history and spray regimes may introduce biases with respect to pesticide types and usage patterns. Therefore, while our results are in keeping with previous studies indicating that pesticide usage alters bacterial community structure, future research should aim to validate self-reported data with controlled experiment to strengthen the reliability of findings. One key limitation of this study is that it focused exclusively on cabbage farms,

although other vegetable crops were also cultivated in the study area using similar pesticide regimes. As such, the findings of our study may not fully capture the broader spectrum of pesticide impacts on soil microbial communities across all types of vegetable farming systems. The identification of bacterial taxa was limited by the reference databases available for 16S rRNA sequencing, which resulted in a high proportion of sequences being classified as unknown. Further studies incorporating metagenome shotgun sequencing and functional assays would provide deeper insights into the ecological roles of these unclassified taxa.

## Conclusion

This study provides evidence that the long-term application of pesticides in cabbage agroecosystems significantly alters the composition and structure of bacterial communities. Consistent with our hypothesis, the richness and diversity of most of the bacteria taxa, and overall abundance were greatly reduced in the pesticide-contaminated soils in relation to the non-contaminated forest soil. The most pronounced decline occurred in the actively treated soils, showing a strong negative effect of ongoing pesticide use. Although some bacterial taxa including *Nitrospira calida*, *Cystobacter* sp., *Pedomicrobium australicum* and *Byssovorax cruenta* increased in abundance within the contaminated soils, possibly due to their adaptive traits or pesticide-degradation abilities, several ecologically important taxa such as *Bacillus* sp. and *Pseudomonas veronii* were significantly reduced. These shifts may impair essential soil processes such as nutrient cycling and organic matter decomposition, with implications for long-term soil fertility and productivity. Interestingly, the bacterial community structure in the abandoned farms showed signs of partial recovery, supporting the hypothesis that cessation of pesticide application may facilitate microbial community recovery. However, the low similarity in bacteria taxa composition between the non-contaminated and abandoned soils suggests that full recovery may take a longer period to occur.

In view of the negative effects of pesticide application on some soil bacteria, our findings highlight the need for more sustainable pest management strategies such as integrated pest management and microbial bioremediation using resilient taxa such as *Rhodoplanes* and *Gemmata* spp. Future research should focus on functional metagenomics to elucidate the roles of the unclassified bacterial taxa and validate their potential for pesticide degradation and agroecosystem support. This study also emphasis the importance of expanding metagenomic assessments across diverse agricultural landscapes in Ghana and beyond.

## Supporting information

**S1 Table. Taxonomic hierarchy of bacteria species within the non-contaminated (NCS), abandoned pesticide-contaminated (AB-PCS) and actively pesticide-contaminated (AC-PCS) soils.**
(DOCX)

**S2 Table. Taxonomic hierarchy of bacteria genus within the non-contaminated (NCS), abandoned pesticide-contaminated (AB-PCS) and actively pesticide-contaminated (AC-PCS) soils.**
(DOCX)

**S3 Table: Taxonomic hierarchy of bacteria family within the non-contaminated (NCS), abandoned pesticide-contaminated (AB-PCS) and actively pesticide-contaminated (AC-PCS) soils.**
(DOCX)

**S4 Table. Taxonomic hierarchy of bacteria order within the non-contaminated (NCS), abandoned pesticide-contaminated (AB-PCS) and actively pesticide-contaminated (AC-PCS) soils.**
(DOCX)

**S5 Table. Taxonomic hierarchy of bacteria class within the non-contaminated (NCS), abandoned pesticide-contaminated (AB-PCS) and actively pesticide-contaminated (AC-PCS) soils.**
(DOCX)

**S6 Table. Taxonomic hierarchy of bacteria phyla within the non-contaminated (NCS), abandoned pesticide-contaminated (AB-PCS) and actively pesticide-contaminated (AC-PCS) soils.**
(DOCX)

## Acknowledgments

We acknowledge all the technicians in the Microbiology Laboratory at the Department of Theoretical and Applied Biology, KNUST, Kumasi, Ghana. We also acknowledge the Soil Research Institute of the Council for Scientific and Industrial Research (CSIR), Kumasi Kwadaso, Ghana. We would also like to thank the Teaching Assistants in the Department of Theoretical and Applied Biology, KNUST for their support. The support of the Adeito and Adakabunso farming community in the Bosome Freho District of Ghana is also well acknowledged.

## Author contributions

**Conceptualization:** Peprah Sefa, Patrick Addo-Fordjour, Bernard Fei-Baffoe, James Damsere-Derry.

**Data curation:** Peprah Sefa.

**Formal analysis:** Peprah Sefa, Patrick Addo-Fordjour, Kwadwo Boampong, James Damsere-Derry.

**Resources:** Peprah Sefa.

**Supervision:** Patrick Addo-Fordjour, Bernard Fei-Baffoe, Kwadwo Boampong, Silas Wintuma Avicor.

**Writing – original draft:** Peprah Sefa.

**Writing – review & editing:** Peprah Sefa, Patrick Addo-Fordjour, Bernard Fei-Baffoe, Kwadwo Boampong, Silas Wintuma Avicor, James Damsere-Derry.

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
