## [Decision Letter · Decision Letter 0]

10 Dec 2024

PONE-D-24-54549Effects of Pesticide Application on Soil Bacteria Community Structure in a Forest Agroecosystem in GhanaPLOS ONE

Dear Dr. Sefa,

Thank you for submitting your manuscript to PLOS ONE. After careful consideration, we feel that it has merit but does not fully meet PLOS ONE’s publication criteria as it currently stands. Therefore, we invite you to submit a revised version of the manuscript that addresses the points raised during the review process.

We look forward to receiving your revised manuscript.

Kind regards,

Shouke Zhang

Academic Editor

PLOS ONE

Journal requirements: When submitting your revision, we need you to address these additional requirements. 1. Please ensure that your manuscript meets PLOS ONE's style requirements, including those for file naming. The PLOS ONE style templates can be found at https://journals.plos.org/plosone/s/file?id=wjVg/PLOSOne_formatting_sample_main_body.pdf and https://journals.plos.org/plosone/s/file?id=ba62/PLOSOne_formatting_sample_title_authors_affiliations.pdf 2. PLOS requires an ORCID iD for the corresponding author in Editorial Manager on papers submitted after December 6th, 2016. Please ensure that you have an ORCID iD and that it is validated in Editorial Manager. To do this, go to ‘Update my Information’ (in the upper left-hand corner of the main menu), and click on the Fetch/Validate link next to the ORCID field. This will take you to the ORCID site and allow you to create a new iD or authenticate a pre-existing iD in Editorial Manager. 3. In your Methods section, please provide additional information regarding the permits you obtained for the work. Please ensure you have included the full name of the authority that approved the field site access and, if no permits were required, a brief statement explaining why 4. We note that the grant information you provided in the ‘Funding Information’ and ‘Financial Disclosure’ sections do not match.  When you resubmit, please ensure that you provide the correct grant numbers for the awards you received for your study in the ‘Funding Information’ section. 5. Thank you for stating the following in your Competing Interests section:  [None]. Please complete your Competing Interests on the online submission form to state any Competing Interests. If you have no competing interests, please state ""The authors have declared that no competing interests exist."", as detailed online in our guide for authors at http://journals.plos.org/plosone/s/submit-now  This information should be included in your cover letter; we will change the online submission form on your behalf.

Reviewers' comments:

Reviewer's Responses to Questions

**Comments to the Author**

1. Is the manuscript technically sound, and do the data support the conclusions?

Reviewer #1: Partly

Reviewer #2: Yes

Reviewer #3: Partly

2. Has the statistical analysis been performed appropriately and rigorously? 

Reviewer #1: No

Reviewer #2: Yes

Reviewer #3: Yes

3. Have the authors made all data underlying the findings in their manuscript fully available?

Reviewer #1: No

Reviewer #2: Yes

Reviewer #3: Yes

4. Is the manuscript presented in an intelligible fashion and written in standard English?

Reviewer #1: Yes

Reviewer #2: Yes

Reviewer #3: Yes

5. Review Comments to the Author

Reviewer #1: The manuscript investigates the effects of pesticides on soil bacterial communities, a highly relevant topic with potential ecological and agricultural implications. While the study employs rigorous methods and provides valuable data, certain sections of the manuscript require refinement to improve clarity, focus, and scientific rigor. In particular, the abstract, introduction, and discussion need to better highlight the key findings and connect them to the broader context of pesticide impacts on soil ecosystems. Additionally, improvements to the sampling design, statistical analysis, and the handling of limitations would further enhance the robustness and interpretability of the research.

Abstract: The abstract presents an important research question but lacks conciseness and clarity. The key results, particularly those related to pesticide impacts on bacterial community diversity, need to be more explicitly highlighted. Additionally, the connections between changes in soil physicochemical properties (such as cation exchange capacity and pH) and bacterial community shifts are not sufficiently addressed. To improve, the abstract should be rewritten to emphasize the major findings and their significance, while also briefly linking the changes in soil properties to microbial shifts.

Introduction: The introduction provides comprehensive background information but is somewhat lengthy and contains sections that do not directly relate to the study's core focus on soil microbiomes. The discussion of pesticide impacts on air, water, and soil is informative, yet it would be more effective if directly tied to soil microbial health. Furthermore, although the study objectives are mentioned, there is a lack of a clear hypothesis or specific research questions. To improve this section, I recommend streamlining the background, focusing more on the direct relevance to soil microbiology, and defining specific hypotheses. Additionally, strengthening the logical flow of language and providing clearer connections between the research objectives and methodology will improve the readability and focus of the introduction.

Study Area: The description of the study area's climate and soil characteristics is adequate, but the manuscript could benefit from more detailed information about the long-term environmental impacts of cabbage cultivation. Specifically, the types of pesticides used, historical application rates, and changes in cultivation area over time are not provided. These details would help readers better understand the context and significance of the findings, especially in terms of pesticide exposure dynamics.

Sampling Procedure: The sampling procedure relies on grouping sampling sites based on farmers' reported pesticide usage history. While this approach is valid, it may introduce some bias, especially if farmers' recollections or records are inaccurate. Furthermore, the "non-contaminated" soil samples were exclusively collected from pristine forest reserves, which may not fully represent agricultural soils that have not been exposed to pesticides. To improve generalizability, I suggest expanding the sampling strategy to include agricultural soils known to have minimal pesticide exposure, providing a more accurate comparison.

Statistical Data Interpretation: The study uses PAST and STATA for data analysis, but the manuscript does not elaborate on advanced ecological statistical methods such as network analysis or redundancy analysis (RDA). These methods could offer deeper insights into the ecological drivers behind microbial community shifts, such as the role of soil physicochemical properties or specific pesticide types. Including a brief explanation of these additional methods, or incorporating them into the analysis, would significantly enhance the robustness of the data interpretation.

Discussion: The discussion addresses the reduced abundance of Pseudomonas and Bacillus in pesticide-contaminated soils, which contradicts previous studies. However, the manuscript does not offer a clear explanation for this discrepancy, and further exploration into possible reasons (such as pesticide type or soil conditions) is needed. Additionally, some sentences are overly complex and densely packed with information, which could hinder readability. For example, the sentence, "Pseudomonas spp. are facultative anaerobes involved in several important processes..." could be simplified and split into more digestible points. It is also recommended to include a section discussing the study's limitations, such as potential biases in sampling or the sequencing method, and suggesting directions for future research. Addressing these limitations would provide readers with a clearer understanding of the study's scope and potential areas for further exploration.

Reviewer #2: COMMENTS TO THE AUTHOR(S):

The manuscript (PONE-D-24-54549) discussed relationship between pesticides and soil microorganisms, which is concerned about falls in the scope of PLOS ONE. However, the quality of the manuscript needs to be further improved.

1.Abstract needs to modify: the abstract should contain Objectives, Methods/Analysis, Findings, and Novelty /Improvement. In Abstract part (Page 3), Authors should not simply describe soil physical and chemical properties, but should explain innovative and main findings.

2.In Page 6, cabbage should add the Latin name.

3.The introduction is too loosely, written to fail to state explicitly the research questions or hypotheses the study aimed to address. What's more, the necessity and innovation of the article should be presented to the introduction.

4.In Page 7, Crops should be supplemented with their Latin names.

5.In Page 8, more details about abandoned pesticide-contaminated and active pesticide-contaminated should be added.

6.In Page 8, “the non-contaminated soil samples were taken from pristine forest reserves ”, Differences in soil microbial composition between forests and farms have been mentioned in several studies. Is it appropriate to treat forest soil as non-contaminated soil?

7.The Discussion part was weakest element of the paper. There was no real discussion in terms of the reliability of the results. There was also a lack of discussion that really positioned the study and its findings within the context of other similar studies. References were strategically placed but the academic discussion around these was missing. As a result, it was difficult to see the contribution that this study makes to development and advancement of knowledge in this area.

8.The conclusion part is mostly qualitative description and lacks quantitative data.

Reviewer #3: The study provides important information on the impact of pesticides on soil bacterial communities and has academic value. However, there is still room for improvement in the experimental design, results analysis, and discussion sections. It is recommended that the authors revise and improve the article based on the reviewers' comments.

1. The study considered three soil samples, so why were the non-contaminated soil samples taken from a pristine forest reserve rather than soil where crops were grown but no pesticides were applied?

2. The article mentions the use of neonicotinoids, bacilli (bio/microbial), and pyrrole pesticides, but does not specify the exact types and concentrations used. It is recommended to provide this information for a more precise understanding of the study's findings.

3. The article only used the Fisher's alpha diversity index, and it is recommended to include additional indices such as the Shannon-Wiener index and Simpson index to provide a more comprehensive assessment of bacterial diversity.

4. The article did not predict bacterial functions, and it is recommended to use databases or functional gene analysis to predict the functions of the bacterial community and explore their potential impact on soil ecosystem functions.

5. The article did not delve into the deeper mechanisms by which pesticides affect bacterial communities. It is recommended to synthesize the results of related studies to analyze the effects of pesticides on bacterial metabolism, growth and reproduction.

6. The overall language of the article is fluent, but some sentences are structurally complex. It is recommended to simplify the language to improve readability.

7. The reference format of the article needs to be checked.

6. PLOS authors have the option to publish the peer review history of their article (what does this mean? ). If published, this will include your full peer review and any attached files.

**Do you want your identity to be public for this peer review?** For information about this choice, including consent withdrawal, please see our Privacy Policy .

Reviewer #1: No

Reviewer #2: No

Reviewer #3: No

---

## [Author Response · Author response to Decision Letter 1]

24 Jan 2025

We have prepared and attached the reviewers comment

---

## [Decision Letter · Decision Letter 1]

13 Feb 2025

PONE-D-24-54549R1Effects of Pesticide Application on Soil Bacteria Community Structure in a Forest Agroecosystem in GhanaPLOS ONE

Dear Dr. Sefa,

Thank you for submitting your manuscript to PLOS ONE. After careful consideration, we feel that it has merit but does not fully meet PLOS ONE’s publication criteria as it currently stands. Therefore, we invite you to submit a revised version of the manuscript that addresses the points raised during the review process.

We look forward to receiving your revised manuscript.

Kind regards,

Shouke Zhang

Academic Editor

PLOS ONE

Journal Requirements:

Reviewers' comments:

Reviewer's Responses to Questions

**Comments to the Author**

1. If the authors have adequately addressed your comments raised in a previous round of review and you feel that this manuscript is now acceptable for publication, you may indicate that here to bypass the “Comments to the Author” section, enter your conflict of interest statement in the “Confidential to Editor” section, and submit your "Accept" recommendation.

Reviewer #1: All comments have been addressed

Reviewer #2: (No Response)

Reviewer #3: All comments have been addressed

2. Is the manuscript technically sound, and do the data support the conclusions?

Reviewer #1: Yes

Reviewer #2: Yes

Reviewer #3: Yes

3. Has the statistical analysis been performed appropriately and rigorously? 

Reviewer #1: Yes

Reviewer #2: Yes

Reviewer #3: Yes

4. Have the authors made all data underlying the findings in their manuscript fully available?

Reviewer #1: Yes

Reviewer #2: Yes

Reviewer #3: Yes

5. Is the manuscript presented in an intelligible fashion and written in standard English?

Reviewer #1: Yes

Reviewer #2: Yes

Reviewer #3: Yes

6. Review Comments to the Author

Reviewer #1: The author has effectively responded to the reviewers' comments, making substantial revisions to clarify the impact of pesticide application on the soil bacterial community structure in a forest agroecosystem in Ghana. These thoughtful changes have notably improved the manuscript’s clarity and scientific rigor.

Reviewer #2: Compared to the first manuscript (PONE-D-24-54549), the quality of this manuscript (PONE-D-24-54549R1) has been greatly improved. However, there are still some problems need to be modified before published.

1.Abstract: “The applied pesticides also caused significant shift in the pH of the examined soils.” significant shift? What is P value? And more data should be added in Abstract section.

2.In Page 4, “Annually, an estimated two million tons of pesticides are applied on agricultural lands worldwide (7). However, in 2022, the total global pesticide use in agriculture was 3.70 million metric tons (Mt) and a total pesticide trade reaching approximately 7.2 Mt of formulated products amounting to USD 41.1 billion (8).” This sentence is confusing to read and it is suggested that the author rewrite this sentence.

3.In Page 12, a low potassium (K) concentration, total potassium or available potassium?

4.In Discussion section, it is suggested to add subheadings for readers' better understanding.

5.The Discussion and Limitation can be combined into Discussion section, and the Conclusion section should be at the end of the manuscript.

Reviewer #3: (No Response)

7. PLOS authors have the option to publish the peer review history of their article (what does this mean? ). If published, this will include your full peer review and any attached files.

**Do you want your identity to be public for this peer review?** For information about this choice, including consent withdrawal, please see our Privacy Policy .

Reviewer #1: No

Reviewer #2: No

Reviewer #3: No

---

## [Author Response · Author response to Decision Letter 2]

15 Apr 2025

RESPONSES TO COMMENTS OF REVIEWER

Comment

Abstract: “The applied pesticides also caused significant shift in the pH of the examined soils.” significant shift? What is P value? And more data should be added in Abstract section.

Response

Please, that statement was wrong as there is no data to suggest shifts in pH. Consequently, we have deleted it from the abstract. We have added more data to the abstract.

Comment

In Page 4, “Annually, an estimated two million tons of pesticides are applied on agricultural lands worldwide (7). However, in 2022, the total global pesticide use in agriculture was 3.70 million metric tons (Mt) and a total pesticide trade reaching approximately 7.2 Mt of formulated products amounting to USD 41.1 billion (8).” This sentence is confusing to read and it is suggested that the author rewrite this sentence.

Response

Please, we have deleted this sentence from the manuscript to maintain clarity.

Comment

In Page 12, a low potassium (K) concentration, total potassium or available potassium?

Response

Please, it is total potassium. This has been clarified in the manuscript (please see red font text on page 9)

Comment

In Discussion section, it is suggested to add subheadings for readers' better understanding.

Response

We have provided subheadings in the discussion section.

Comment

The Discussion and Limitation can be combined into Discussion section, and the Conclusion section should be at the end of the manuscript.

Response

We have combined the discussion and the limitations, but because we sub-headed the discussion, we have provided a subheading for the limitations under Discussion.

We have placed the conclusion at the end of the manuscript.

---

## [Editor Report · Decision Letter 2]

17 Apr 2025

Effects of Pesticide Application on Soil Bacteria Community Structure in a Forest Agroecosystem in Ghana

PONE-D-24-54549R2

Dear Dr. Peprah Sefa,

We’re pleased to inform you that your manuscript has been judged scientifically suitable for publication and will be formally accepted for publication once it meets all outstanding technical requirements.

Kind regards,

Shouke Zhang

Academic Editor

PLOS ONE
---

## [Editor Report · Acceptance letter]

PONE-D-24-54549R2

PLOS ONE

Dear Dr. Sefa,

I'm pleased to inform you that your manuscript has been deemed suitable for publication in PLOS ONE. Congratulations! Your manuscript is now being handed over to our production team.

Kind regards,

on behalf of

Dr. Shouke Zhang

Academic Editor

PLOS ONE